# Crosslinking of Gelatin in Bicomponent Electrospun Fibers

**DOI:** 10.3390/ma14123391

**Published:** 2021-06-18

**Authors:** Judyta Dulnik, Paweł Sajkiewicz

**Affiliations:** Laboratory of Polymers and Biomaterials, Institute of Fundamental Technological Research, Polish Academy of Sciences, Pawińskiego 5b, 02-106 Warsaw, Poland; jdulnik@ippt.pan.pl

**Keywords:** crosslinking, gelatin, nanofibers, biodegradable polymers, electrospinning

## Abstract

Four chemical crosslinking methods were used in order to prevent gelatin leaching in an aqueous environment, from bicomponent polycaprolactone/gelatin (PCL/Gt) nanofibers electrospun from an alternative solvent system. A range of different concentrations and reaction times were employed to compare genipin, 1-(3-dimethylaminopropyl)-N’-ethylcarbodimide hydrochloride/N-hydroxysuccinimide (EDC/NHS), 1,4-butanediol diglycidyl ether (BDDGE), and transglutaminase. The objective was to optimize and find the most effective method in terms of reaction time and solution concentration, that at the same time provides satisfactory gelatin crosslinking degree and ensures good morphology of the fibers, even after 24 h in aqueous medium in 37 °C. The series of experiments demonstrated that, out of the four compared crosslinking methods, EDC/NHS was able to yield satisfactory results with the lowest concentrations and the shortest reaction times.

## 1. Introduction

One of the trends that has been observed for years in tissue engineering is an attempt to mimic mother nature. Many of the approaches can seem revelatory, but until the human body accepts our invention as something familiar and cells recognize the environment we prepared for them as suitable, we are going to miss our goal.

Extracellular matrix (ECM), apart from cells, is a key ingredient that constitutes almost every tissue. In many cases, ECM is a highly fibrillar structure and matching this characteristic can be crucial to the success of our approach. This makes electrospinning, though not a new idea itself, a technique that has still a lot to offer, considering a remarkable structural similarity of electrospun materials to ECM. Electrospinning is a simple and flexible method that enables obtaining nanofibers from an array of polymers with high porosity and tunable properties [1,2,3]. 

Previously, we optimized the method of obtaining bicomponent nanofibers made of polycaprolactone (PCL) with an addition of gelatin, through electrospinning from a green, cheap, and safe for the operator solvent system—a mixture of acetic and formic acid [4]. Polycaprolactone is a biodegradable aliphatic polyester with good mechanical properties, present in numerous biomedical studies, but it lacks bioactivity that only natural polymers can provide [5,6]. Gelatin, being a derivative of collagen, a protein that is present in abundance in ECM, is a great source of Arg-Gly-Asp (RGD) amino acid sequences that stimulate cell attachment [7,8,9]. 

Unfortunately, as our previous studies have shown, gelatin, being soluble in aqueous conditions is prone to be leached from the fibers, which decreases materials’ bioactive potential [10]. The solution to this predicament is gelatin crosslinking within the fiber. While various physical methods such as ultraviolet irradiation, dehydrothermal treatment [11,12,13], as well as chemical ones are used to stabilize scaffold materials made of gelatin, only some of them are suitable in case of PCL/Gt nanofibers. 

In this work, we decided to focus on chemical methods and investigate a set of different crosslinking agents. The main concerns were that the methods were of low toxicity and had innovative potential. While the most widespread and best studied chemical agent is glutaraldehyde, there are works that indicate its detrimental effect on the cellular response and because of that it was not taken into consideration [14,15,16]. Based on the criteria stated above, four crosslinking agents were chosen: genipin, 1-ethyl-3- (3-dimethylaminopropyl) carbodiimide hydrochloride/N-hydroxysuccinimide (EDC/NHS), 1,4-butanediol diglycidylether (BDDGE), and transglutaminase. 

Genipin is a crosslinking agent of plant origin, considered to be of low toxicity, widely used with natural polymers like chitosan and collagen, reacting with primary amine groups such as lysine and arginine [17,18]. EDC/NHS is a highly efficient method, popular with biomaterials aimed at tissue engineering applications. It is a zero length crosslinker, which forms amide bonds between carboxyl and amine groups, but does not take part in newly formed linkage [19,20,21]. BDDGE is a widely used crosslinker in cosmetology with hyaluronic acid based dermal fillers. It is biodegradable and of low toxicity [22,23]. Transglutaminase is an enzyme that has become popular in recent years as a crosslinking agent in scaffolds for tissue engineering. It can be of tissue or bacterial origin and it is confirmed to improve adhesion and spreading of cells in-vitro [24,25]. Transglutaminase has a beneficial effect on wound healing, as it is an enzyme that is naturally secreted by fibroblasts in response to tissue injury [26]. Both BDDGE and transglutaminase form a bond between two amine groups. 

The intention of this work was to establish optimal conditions of gelatin crosslinking within PCL/Gt fibers that ensure a relatively high gelatin amount would still remain in the material even after being subjected to an aqueous environment and that nanofibers’ morphology would not be compromised in the process. Such perspective is innovative considering previous results related entirely to crosslinking of pure gelatin fibers.

## 2. Materials and Methods

### 2.1. Materials

PCL (Mn = 80 kDa), gelatin from porcine skin, type A (gel strength 300), phosphate buffer saline tablets and formic acid (98%) were purchased from Sigma-Aldrich, St. Louis, MO, United States. Glacial acetic acid (99.5–99.9%) was purchased from POCH, Gliwice, Poland. Ethanol (99.8%) was purchased from Chempur, Piekary Śląskie, Poland. EDC and NHS were purchased from Thermo Scientific, Waltham, MA, United States. Genipin was purchased from Challenge Bioproducts Co., Ltd., Touliu, Taiwan. Transglutaminase (1490 U/g) was purchased from P.M.T. Trading, Łódź, Poland.

### 2.2. Electrospinning

Polymer solution for electrospinning was prepared by dissolving PCL and gelatin separately in a mixture of acetic and formic acid (9:1 *w/w* ratio) with *w/w* 15% concentration by stirring for 24 h. Both solutions were then mixed together in a 7:3 ratio (70% PCL, 30% gelatin) and stirred for an additional 2 h. The prepared solution was electrospun on a grounded rotating drum collector with surface linear velocity 0.4 m/s, which ensures no fiber parallelization. The distance from the needle to collector surface was set to 15 cm, flow rate was 1 mL/h, and the voltage used varied from 12 to 14 kV. Humidity was kept at 45–50%.

### 2.3. Crosslinking Methods

Electrospun material sheets were cut into 1.5 cm × 4.5 cm samples. Before crosslinking, they were placed in a vacuum oven for not less than 24 h to remove any moisture that gelatin within the fibers is prone to attract. Immediately after taking samples out of the vacuum oven, the weight of each sample was measured. Since one of the ways of determining crosslinking degree is calculating gelatin mass change after fiber biodegradation test, this step is important because of the varying ambient humidity in the laboratory. 

A series of preliminary studies was performed for all crosslinking methods. It consisted of a “trial and error” approach that led, for each of the compounds, to establishing a set of conditions to be compared in the main part of the work, as well as an optimal solvent composition. The ranges of the crosslinking reaction conditions that were chosen for the final comparison can be found in Table 1. For EDC/NHS, a molar ratio of 2:1 was kept constant for all tested concentrations. 

Every crosslinking experiment was performed on three samples. After each experiment was concluded, samples were thoroughly rinsed for 1 h, while vigorously stirred. In the preliminary step, demineralized water was used as a rinsing medium, but for the final comparison experiments we decided to use the same type of solvent that crosslinking agent was dissolved in instead. The explanation for this change can be found in the Section 3.1. The rinsed samples were placed in a vacuum oven for not less than 24 h and then their weight was measured again.

### 2.4. Fibre Biodegradation Test

All crosslinked samples, as well as non-crosslinked control, were placed for 24 h in phosphate buffer saline (PBS) solution, with an addition of 0.1% sodium azide to prevent bacterial or fungal growth, at 37 °C and gently stirred. Afterwards they were rinsed for 1 h in demineralized water, while vigorously stirred, to remove any PBS salts residues. Rinsed samples were placed in a vacuum oven for not less than 24 h and their weight was measured once again.

A large part of this discussion will be based on gelatin content, or mass, within the material expressed in ‘%’. These values were concluded from sample weight change that occurs during both crosslinking reaction and subsequent 24 h of biodegradation test. To be able to calculate gelatin mass after any of those steps, an assumption was made that of the two polymers present in the material, any weight decrease will be assigned solely to gelatin. A time range for any weight loss to occur for polycaprolactone, whether it is molecular or absolute, varies from 6 to 24 months [10]. It can be safely said that any material weight change that occurs within the 24 h of immersion in PBS solution has to be related to gelatin leaching. So, the term biodegradation used throughout the text for immersion in PBS solution test has a context of degradation of the fiber structure as a result of gelatin leaching.

The material used in this work consisted of 30% gelatin and 70% polycaprolactone. The recorded change in a sample mass (Δm) was, as reasoned above, attributed to gelatin loss. Gelatin content after biodegradation test ($Gt_{C1}$) reported in this work was calculated for each sample separately, in relation to a 100% of the sample’s gelatin initial mass ($m_{Gt}$) following the equations: (1)$$m_{Gt}+\Delta m=m_{Gt1}$$
(2)$$\frac{m_{Gt1}}{m_{Gt}}*100\%=Gt_{C1}$$
where $m_{Gt1}$ was gelatin mass after biodegradation test.

### 2.5. Scanning Electron Microscope (SEM) Imaging

SEM imaging was performed with JEOL JSM6010LV, Japan, Tokio. The samples were examined after crosslinking as well as after biodegradation test. To ensure good conductivity, samples were sputter coated with 10 nm of gold, using JEOL smart coater.

### 2.6. Fourier Transform Infrared (FT-IR) Spectroscopy

The FT-IR measurements of the crosslinked samples after biodegradation test as well as non-crosslinked control was performed with FT-IR spectroscopy with attenuated total reflection (ATR) technique using VERTEX 70, Bruker, Billerica, MA, United States. 

## 3. Results and Discussion

### 3.1. Solvent Optimisation

Apart from transglutaminase that only dissolves in water, the three other crosslinking agents chosen for this research are commonly found in the literature dissolved in ethanol or ethanol/water mixtures [17,19,21,22]. A series of preliminary crosslinking experiments were performed to establish (if possible) one type of solvent for these three compounds. 

The first crosslinking agent that underwent solvent optimization was genipin. The experiments were started with 5% concentration over 24 and 48 h with solvent mixtures of ethanol/demineralized water with the latter content of 10, 20, and 30%. Genipin crosslinking is known for the color change that occurs during the course of the reaction [18]. A gradual color transition of the mat, from white to the shades of blue can be observed in Figure 1. Color blue intensity strongly correlated with increasing water content of used solvent and mildly with the reaction time.

The change in gelatin content showed that with the increased water concentration in the solvent, the more gelatin content was left in the samples. It was true both for the samples weighted after only crosslinking, as well as after the biodegradation test, that followed crosslinking (Figure 2). While the gelatin content of the samples after crosslinking for 24 h and 48 h differed for only 3–6%, a significant difference was observed in the comparison done after biodegradation test. These results showed the need for experiments with different concentrations and reaction times in order to find optimal ethanol to water ratio.

A lower genipin concentration of 3% was used next with reaction time set to 48 h and solutions with water content ranging from 20 to 50%. The results showed that the samples crosslinked with 3% genipin over 48 h reached slightly higher gelatin content, after biodegradation test, than those crosslinked with 5% for 24 h (Figure 2), as well as a significant decrease of gelatin content for samples with more than 30% water in crosslinking solution.

The reduction of gelatin mass for samples crosslinked with solutions containing more than 30% of water can be explained by domination of gelatin dissolution over the crosslinking process. On the other hand, too little water content in solution was insufficient for effective crosslinking, resulting in maximum crosslinking efficiency at around 30% of water in solution. We suppose, that for those samples, the drop in gelatin content after only being subjected to crosslinking can be explained by gelatin dissolving during rinsing for 1 h in demineralized water. For that reason, it was decided that the rinsing medium for all tests from that moment forward would be the same as the solvent used during crosslinking. 

A series of EDC/NHS crosslinking tests with increasing water content in the solvent was performed next, in order to compare the results with the observations for crosslinking with genipin. 

In the case of EDC/NHS crosslinking (Figure 3), the same trend was confirmed, where gelatin mass measured for samples after biodegradation test was the highest for crosslinking solution with 30% water and 70% ethanol. It was proved that the revision of the rinsing medium resulted in a prevention of gelatin loss during rinsing after crosslinking, that was observed for genipin tests with water content in crosslinking solutions less than 30%. Similarly to the results of tests with genipin, there is an evident decrease in gelatin content after biodegradation for the samples with less than 30% of water in crosslinking solution. The lowest value was recorded for 100% ethanol solvent solution, being almost the same as for the control non-crosslinked samples. It suggests that no, or extremely limited, crosslinking occurred in that solution and that water presence is crucial for the reaction to occur. Preliminary tests done with BDDGE also confirmed the same trend (data not shown) for solutions with water content below 30%. The question regarding the nature of this observation is valid and should be studied further.

Although gelatin is soluble in water at and above 37 °C, our experience showed that with nanofibrous materials it readily dissolves at lower temperatures. Because of the very large surface area that such materials have, makes it much easier for gelatin to dissolve [27]. Taking this into consideration, we wanted to find a solvent that will make it possible for crosslinking to happen, while not sabotaging the experiment by dissolving gelatin before the reaction can take place. After conducting these series of experiments, we decided to use ethanol/water mixture with a 7:3 *w/w* ratio for all tests planned in the final step (apart from transglutaminase).

### 3.2. Gelatin Mass Change

To be able to compare crosslinking efficiency in preserving gelatin within the fiber by those four methods, it was decided to arbitrarily define a value of gelatin content, after biodegradation test, that we consider a positive result. We deemed every set of gelatin crosslinking conditions that resulted in at least 85% gelatin retention to be satisfactory and successful. 

The only crosslinking agent that failed to reach this value was transglutaminase. Although the samples crosslinked for 24 h with the highest transglutaminase concentration preserved 87% of gelatin, this result did not remain as high after biodegradation (Figure 4). Since transglutaminase is not soluble in organic solvents, there was no other option than to use water as a solvent. As was previously showed, gelatin within the fibers is prone to quick dissolution in aqueous environment, even at room temperature, so it was little to no surprise that the samples after crosslinking for a longer period of time, 72 h, achieved even worse results.

Due to this, transglutaminase was left out of the most comprehensive part of the experiment. The other three crosslinking agents were tested with a vast array of concentrations and reaction times. Gelatin content values for crosslinked samples after biodegradation test are presented in Figure 5a,c,e. To represent zero crosslinking time, there was used a value obtained for control, non-crosslinked samples—for which the gelatin content after biodegradation test was only 27.1%.

From the results of each crosslinking method, a set of points representing crosslinking agent concentration in dependence of time needed to achieve the exact 85% of gelatin content within the fibers, $f\left(C\right)=t_{0.85}$ (Figure 5b,d,f) was calculated. This way it is possible to determine a distinct reaction time for any given concentration of crosslinking agent, that will grant 85% gelatin preserved in the material after 24 h of biodegradation. It is also a cut-off line, above which a satisfactory crosslinking efficiency should be expected every time.

Looking at the genipin crosslinking results (Figure 5a), it can be seen that the values of gelatin content above 85% could be achieved for 6% solution in 24 h, for 4% in 48 h, and for 2% in 72 h. Solutions with 0.5% never reached this level, and 1% needed 168 h. 

As for BDDGE (Figure 5c), there were needed higher solution concentrations to achieve similar results as for genipin. Solution with the concentration of 15% reached the set threshold of gelatin content in 24 h, 10% needed 48 h, and 5% achieved it in 72 h. Again, the lowest 2.5% concentration was not able to successfully crosslink gelatin during 168 h of experiment. In comparison to genipin, for the examined concentrations, there was needed almost thrice as much time to reach an equal level of crosslinking efficiency, or thrice higher concentration, to achieve this result in the same time. 

In the case of EDC/NHS crosslinking method (Figure 5e), even solution with an EDC concentration as low as 0.02% reached 85% gelatin content in 5 h, while 0.23% concentration, which was the highest that was tested, obtained the same result in only 20 min. 

The difference in the rate of reaction of EDC/NHS method in comparison to genipin and BDDGE, as well as the gap between the useful concentrations of EDC/NHS and the other two compounds are great. To be able to compare all of these results in one graph a logarithmic scale needed to be used (Figure 6).

Gathering data from every single crosslinking experiment performed with the use of genipin, EDC/NHS and BDDGE in one graph (Figure 6), can help comprehend how much EDC/NHS method is faster than the other two. It is easy to see that what results can be achieved by EDC/NHS in 1 h corresponds to what is expected from genipin in 10 h, and even more time from BDDGE (Figure 6a). Roughly the same crosslinking efficiency can be obtained for 100 h, lasting experiment by 10% and 1% solutions of BDDGE and genipin, respectively, while we would only need to spend 20 min using 0.1% EDC to get the same result (Figure 6b).

The experimental data of *t*_0.85_ vs. *C* were approximated numerically with exponential function, allowing quantitative comparison of the sensitivity of *t*_0.85_ to the crosslinking agent concentration from the slopes of the double logarithmic plots (Figure 7). It is clear that the rate of *t*_0.85_ reduction with increasing crosslinking agent concentration is the fastest for EDC/NHS. Between the other two, BDDGE has a higher slope value than genipin. It may suggest that theoretically, with much higher reagent concentrations, there is a certain concentration for which both would reach the same values of *t*_0.85_.

We suppose that two characteristics of EDC crosslinking reaction mechanism may possibly contribute to such a great difference in the rate of gelatin crosslinking reaction with EDC/NHS and the other two compounds.

The first is EDC, being the only zero length crosslinker in the tested group. It means that its concentration in the solution does not decrease during the course of the reaction. EDC molecule after facilitating the formation of a bond, does not become a part of this newly formed linkage, but it is released back into the solution and can be used again in crosslinking reaction for an infinite amount of times. Both genipin and BDDGE embed themselves between the two functional groups that they eventually link, and so their concentration does decrease in time.

The second reason is the fact that, again, EDC is the only crosslinking agent here, that helps create a bond between a COOH and a NH_2_ group, when genipin and BDDGE insert themselves between two NH_2_ groups. While each gelatin molecule has numerous NH_2_ groups from lysine and arginine, it also has an even bigger amount of COOH groups from aspartic and glutamic acid [28]. It alone multiplies the number of possible bonds to be formed within for the same amount of gelatin by two. Simultaneously to a reaction occurring between two gelatin molecules, EDC is also linking COOH groups from polycaprolactone and NH_2_ groups from gelatin, creating a strong interface between polycaprolactone and gelatin within the fiber. This may be a factor in prevention of gelatin leaching from the fiber in aqueous environment.

### 3.3. Morphology of the Fibres

One of the criteria of a good crosslinking agent for the bicomponent PCL/Gt nanofibers is that, apart from the ability to prevent gelatin from leaching in aqueous medium, it at the same time does not damage fiber morphology.

The 24 h biodegradation experiments showed that a control PCL/Gt, non-crosslinked sample is left with only 27.1% of its initial gelatin content. The smooth and uniform fibers (Figure 8a) after most of the gelatin addition is left, become eroded, with clearly visible grooves and elongated holes (Figure 8b).

For the purpose of comparing the morphology of fibers after crosslinking, there were selected images presenting only materials of which gelatin content after biodegradation test was not less than 85% (Figure 9). This prerequisite was defined in order to establish crosslinking conditions that meet both morphology and gelatin retention simultaneously. The only exception in this comparison is a picture (Figure 9a,b) of a sample crosslinked with transglutaminase that had a gelatin content after biodegradation test of 74% (which was the best result for this method).

Figure 9 presents SEM images of samples’ morphology after crosslinking with all four compounds and after biodegradation test. The differences between these pairs of images for each crosslinking method are minimal, which shows that the majority of the damage to the fibers’ shape and nonwoven architecture happened during crosslinking process.

While none of the crosslinking methods preserved every aspect of the fibers’ morphology, the results differ in the damage extent. The samples that underwent crosslinking with both EDC/NHS and genipin (Figure 9c–f) maintained, for the most part, the thickness of the fibers. In both cases, fibers were slightly wavy and wrinkled with minimal fusing but overall fiber arrangement was preserved. On the opposite side are the fibers crosslinked with transglutaminase and BDDGE (Figure 9a,b,g,h). In both materials fibers were heavily fused together, the pores irregular in size and in a small number. For transglutaminase, it might be an effect of both insufficiently crosslinked fibers as well as crosslinking reaction taking place in fully water based solution.

### 3.4. Gelatin FT-IR Analysis

FT-IR analysis of gelatin within the crosslinked materials has been focused on amide I and amide II bands being two major bands of the protein infrared spectrum. The amide I band located between 1600 and 1700 cm^−1^ is mainly associated with the C=O stretching vibration, while amide II band in the range 1510–1580 cm^−1^ results from the N-H bending vibration and from the C-N stretching vibration [29]. Both bands are sensitive to the molecular conformation.

Because of a multicomponent nature of both bands and their relatively large width, particularly for amide I, gelatin content determination in investigated samples from the bands intensity was not reliable, particularly in comparison with the weight change measurement. What is evident from FT-IR spectra in Figure 10 is the shift of the maximum absorbance of amide II band toward higher wave numbers in crosslinked samples compared to non-crosslinked material. This shift is an evidence of the crosslinking process, which affects clearly bending vibrations of N-H and/or stretching vibrations of C-N bonds.

## 4. Conclusions

The systematic analysis of gelatin crosslinking within PCL/Gt fibers electrospun from acetic and formic acid solution resulted in optimization for the first time of parameters of the crosslinking process for this type of material. The established process conditions with low reagent concentrations and short reaction times lead to PCL/Gt fibers with high gelatin content and stable morphology after 24 h of biodegradation test.

Taking into account low concentrations of the reagents and short reaction times needed for EDC/NHS to perform a successful reaction, the results presented in this work proved that this method meets all criteria mentioned above best. EDC/NHS was also shown to have a small effect on fibers’ morphology. It is worth noting that as EDC is a zero-length crosslinker and no reagent is present in the material after it rinsing.

Further optimization studies of EDC/NHS crosslinking conditions will be performed on a group of nonwovens with a varying PCL to gelatin ratios, as well as a different solvent used in electrospinning. Crosslinked samples will undergo biodegradation for up to 30 days, uniaxial tensile testing as well as cytotoxicity and cellular response studies.

Maintaining gelatin presence within the materials as well as preserving fibers’ morphology is crucial from the point of view of possible applications of PCL/Gt nonwovens. We believe using EDC/NHS as a crosslinking agent for these bicomponent electrospun nanofibers, elevates greatly their potential in the field of tissue engineering and regenerative medicine therapies.

## Figures and Tables

**Figure 1 materials-14-03391-f001:**
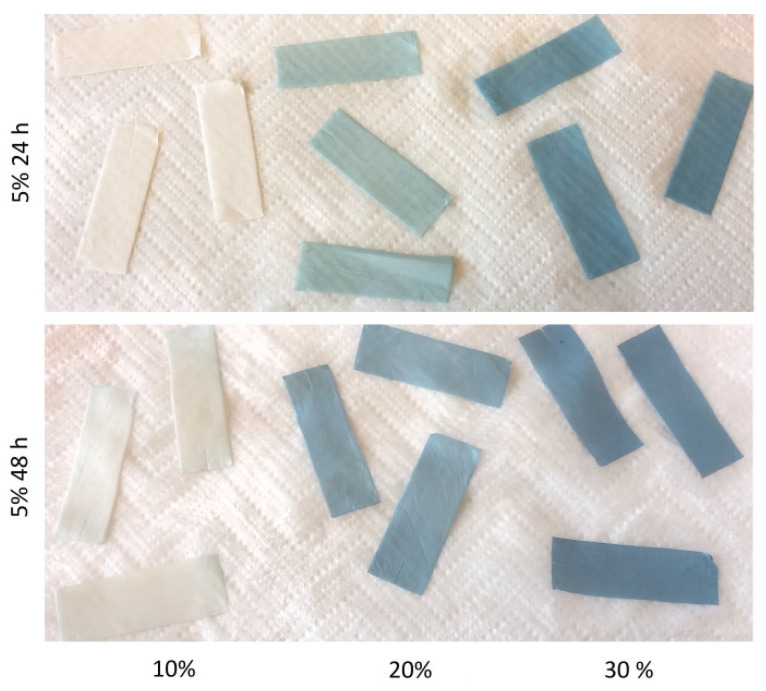
Samples crosslinked with 5% genipin solution for 24 (**up**) and 48 h (**down**) with solvents with increasing water content.

**Figure 2 materials-14-03391-f002:**
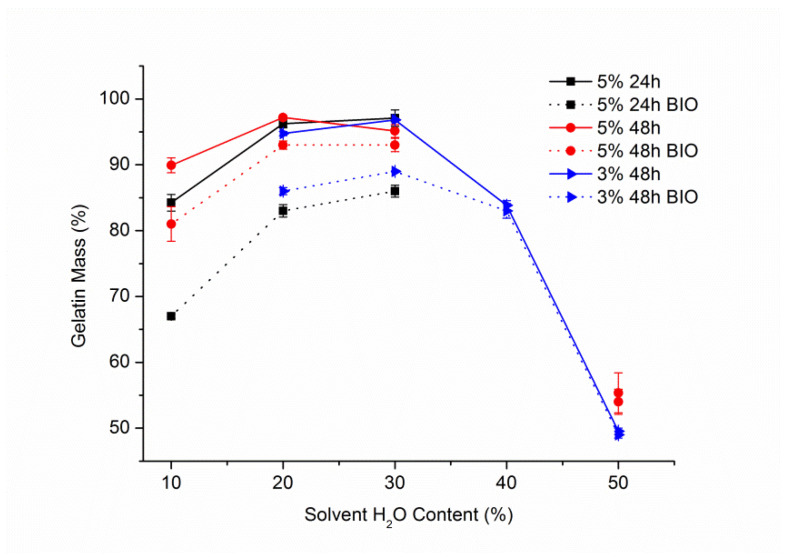
Gelatin content of the samples after crosslinking with genipin solutions and after biodegradation (BIO) test vs. water content in the solvent.

**Figure 3 materials-14-03391-f003:**
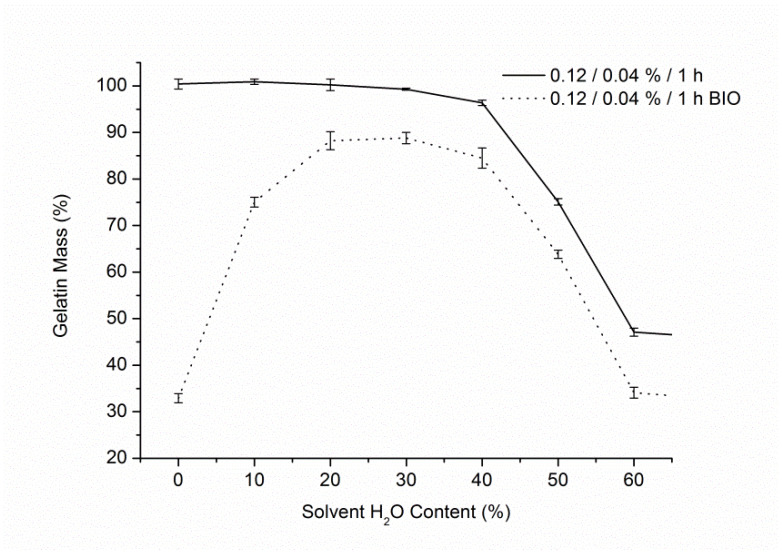
Gelatin content of the samples right after crosslinking with EDC/NHS and after biodegradation test vs. water content in the solvent.

**Figure 4 materials-14-03391-f004:**
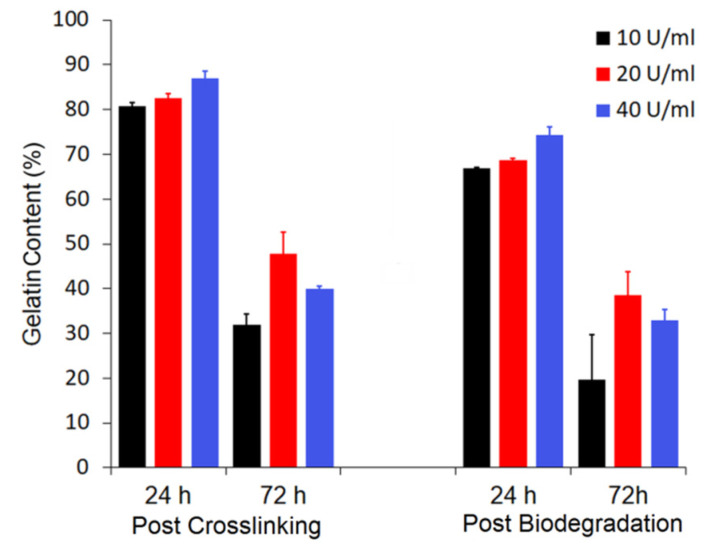
Gelatin content for samples crosslinked with transglutaminase, both after reaction and biodegradation test.

**Figure 5 materials-14-03391-f005:**
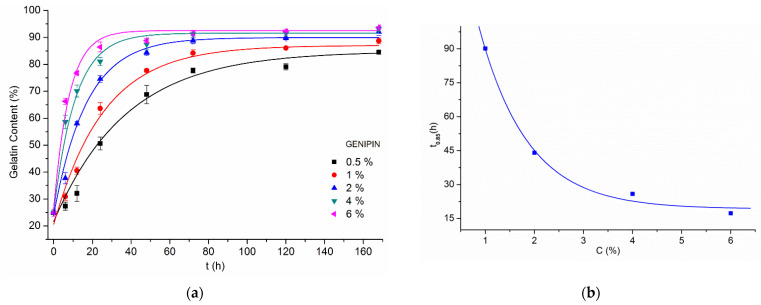
Graphs showing gelatin content after biodegradation test of samples crosslinked with different compounds and with varying conditions (left) and $f\left(C\right)=t_{0.85}$ (right) for: (**a**,**b**) genipin, (**c**,**d**) BDDGE, (**e**,**f**) EDC/NHS. In graph (**f**), *C* refers to EDC concentration. All presented data were fitted with an exponential function.

**Figure 6 materials-14-03391-f006:**
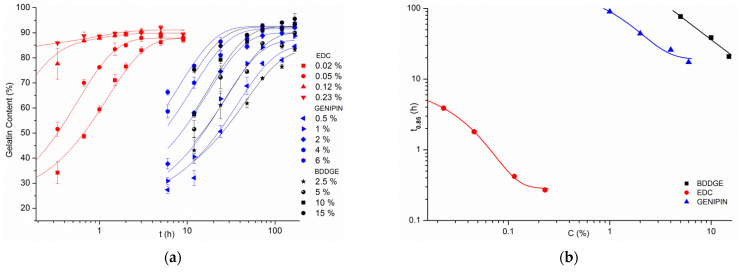
Graphs combining crosslinking results from all experiments: (**a**) gelatin content after biodegradation test, (**b**) $f\left(C\right)=t_{0.85}$ using logarithmic scales. For EDC/NHS, only EDC concentrations are mentioned, in order not to reduce the transparency of the graphs.

**Figure 7 materials-14-03391-f007:**
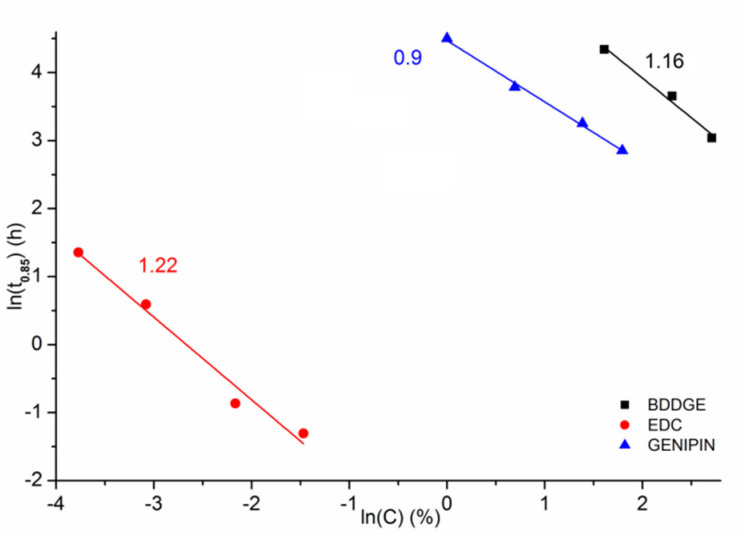
Graph combining *t*_0.85_ vs. *C* for all experiments with slope values.

**Figure 8 materials-14-03391-f008:**
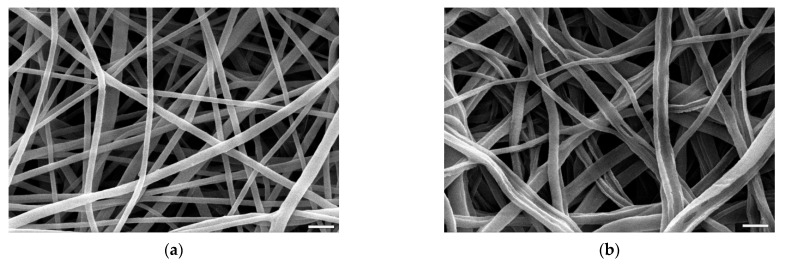
SEM images of control non-crosslinked samples: (**a**) before, (**b**) after biodegradation test. The marker equals 10 µm.

**Figure 9 materials-14-03391-f009:**
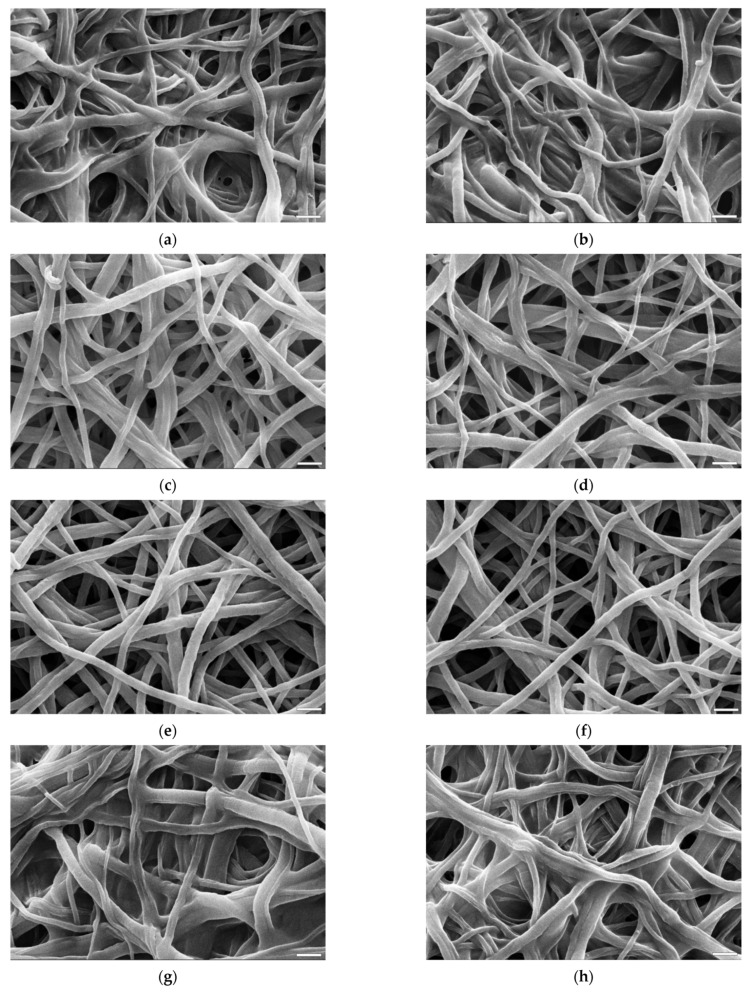
SEM images of samples crosslinked with different methods: before (left) and after (right) 24 h of biodegradation test: (**a**,**b**) transglutaminase 40 U/mL, 24 h, (**c**,**d**) genipin 4%, 48 h, (**e**,**f**) EDC/NHS 0.12%/0.04%, 40 min, (**g**,**h**) BDDGE 10%, 48 h. The marker equals 10 µm.

**Figure 10 materials-14-03391-f010:**
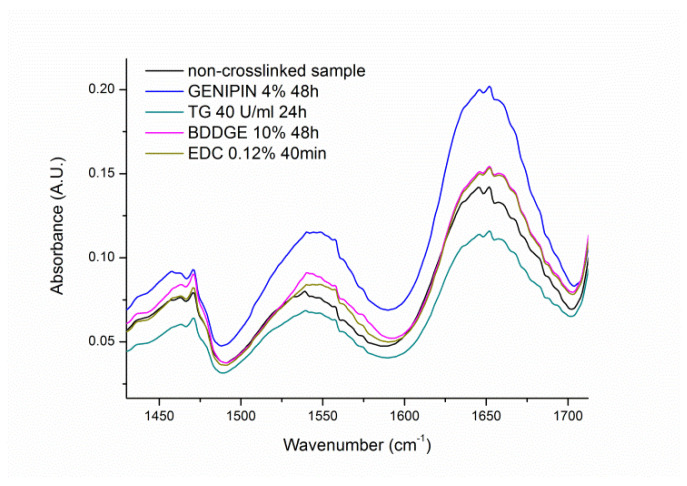
Characteristic FT-IR spectra of PCL/Gt nonwoven samples crosslinked with different methods after biodegradation test (samples in correspondence to SEM analysis). As a control a non-crosslinked sample before biodegradation test was used.

**Table 1 materials-14-03391-t001:** Crosslinking conditions.

Crosslinking Agent	Concentration (% *w/w*)	Process Duration	Solvent (*w/w* Ratio)
Genipin	0.5–6%	6–168 h	EtOH:H_2_O (7:3)
Transglutaminase	10–40 U/mL	24–72 h	H_2_O
EDC/ NHS	0.02–0.23% 0.01–0.08%	10 min–9 h	EtOH:H_2_O (7:3)
BDDGE	2.5–15%	12–168 h	EtOH:H_2_O (7:3)

## Data Availability

Data sharing is not applicable for this article.

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
