# Peer review of "Crosslinking of Gelatin in Bicomponent Electrospun Fibers"

_materials, 2021, doi:10.3390/ma14123391_

Round 1
Reviewer 1 Report
The manuscript attempted to develop a method to cross-link gelatin in PCL/gelatin nano fibers using four different methods, including genipin, EDC/NHS, BDDGE and an enzyme. They found that EDC/NHS is the most efficient method for cross-linking gelatin. Gelatin cross-linking has been extensively studied in the literature. Physical cross-linking methods are very simple to be implemented with great stability which has been well documented including my own group work. Among four chemical methods in the manuscript, genipin, EDC/NHS and BDDGE have been studied in other gelatin nano fibers. The enzyme reaction with a solid surface is expected to have a poor efficiency. The manuscript thus did not have enough novelties, neither new chemical agents nor novel fiber preparation methods. The manuscript is read like a report, and the quality has a lot of room for improvement.
Reviewer 2 Report
Crosslinking of Gelatin in Bicomponent Electrospun Fibres by Judyta Dulnik and Paweł Sajkiewicz
This paper by Dulink and Sajkiewicz discusses the application of four different chemical cross-linking methods targeting gelatin in electrospun poly(caprolactone)/gelatin fibres. The bicomponent fibres were electrospun from a solvent mixture (9/1 w/w acetic/formic acid) that the authors developed previously. Optimisation of cross-linking conditions is discussed based on the change in weight of the fibre sample (attributed to gelatin only) after cross-linking and after a biodegradation test, along with SEM analysis.
An overall well-written report, which however is in my view rather limited in scope in its present form. The authors discuss their otherwise detailed cross-linking optimisation and comparison based only on the weight change attributed to gelatin, and do not present any other data (e.g. FTIR, DSC, TGA) that would have been of value in the same direction. As the vast majority of the paper is focused on the weight change monitoring, there are several occasions (some highlighted in the detailed comments below) where the conclusions discussed could have been reached earlier and in a more direct way (e.g. increased presence of water and extended exposure of the fibres to it leads to enhanced leaching of gelatin). As a result, the SEM analysis is brief and rather superficial.
I believe that the paper has overall merit and could be re-considered after major revision, as long as the authors can present additional data (e.g. FTIR, DSC, TGA) on the four cross-linking approaches (e.g. compare between as-spun fibres/cross-linked fibres/both groups after biodegradation test – also in SEM). This would help in confirming the occurrence/nature/extent of crosslinking in each case and compare between the four variants, selectively reduce the current part on weight change, and expand the SEM analysis. Moreover, the authors tend to use first person syntax quite a lot (we, us, our) and it would be recommended to balance this with more third person syntax.
More detailed comments can be found below.
Abstract
Lines 10-11: remove capitals from 1-(3-Dimethylaminopropyl) and N-Hydroxysuccinimide
Line 12: “…time- and cost-effective….”
Introduction
Lines 21-23: this segment reads informal, please revise
Line 37: full term for acronym RDG is not given
Line 41: space between “potential” and “[40]”
Line 42: extra space after “…irradiation ,”
Line 46: please replace “low toxic” with “of low toxicity” – several occurrences in the text
Line 47: “…had innovation potential.”
Line 51: “ether” is redundant, please remove
Line 52: “…and transglutaminase.”
Line 53: low-toxic, please see above
Line 57: “…does not take part…”
Lines 61-62: “…it is confirmed to improve adhesion and spreading of cells in-vitro.” A citation is needed here.
Line 66: please replace “wanted” with a more formal expression (aimed, intended)
Materials & Methods
Line 83: please remove “thus” and replace with “…the prepared solution…”
Line 90: please rephrase “…get rid of…”
Lines 91-92: please rephrase “Right after taking them out of it,…”
Table 1, column “Concentration”: the concentration expression needs to be defined: is it %w/w, %w/v, other?
The used ratio of EDC to NHS is not clear from the ranges given in Table 1. What was the selected ratio and why, and has it been kept constant? See also line 227.
Lines 104-105: “…for the most part, …” This is vague – please either specify or remove
Line 112: “…at 37°C”
Line 122: probably best to remove “…what was true for our previous studies, …” and just keep the citation [10] as the overall sentence is quite long
Line 127: spaces needed at “…mass(∆?)was…”
Line 134: “SEM imaging was done with JEOL JSM6010LV”, please rephrase/formalise
Results & Discussion
Line 140: “…commonly found in the literature…” – citation(s) needed
Lines 145-146: “Genipin crosslinking is known for the colour change that occurs during the course of the reaction.” – citation(s) needed
Line 157 & 163: please quantify the “…slightly higher gelatin content” from Fig 2
Line 166: remove “right”
Lines 168-170: I would suggest to remove this segment, doesn’t add value in my view
Lines 173-175: “On the other hand, too little water content in solution was insufficient for effective crosslinking resulting in maximum crosslinking efficiency at around 30% of water in solution.”
Why is that? How does this relate to the solubility of the cross-linkers? The authors would need to expand here. Same for lines 190-192.
Line 184: see comment on EDC/NHS ratio in Table 1
Line 189: best to replace “lack of gelatin loss” with “gelatin retention” or similar
Line 198: replace “enormously” with “very”
Line 200: “Having this in mind, we wanted to find…” – please formalise/rephrase
Line 209: “…at least 85 % gelatin content retention”
Line 227: see comments on EDC/NHS ratio, and replace “0” with “zero”
Line 231: “…dependence on time…”
Lines 242-243: please rephrase “get there” and fully revise the last sentence, it is too informal.
Lines 246-247: “15 % barely reached 85 % gelatin content in 24 h, 10 % needed 48 h and 5 % achieved it in 72 h.” Too many percentages referring to both samples and gelatin content – please revise.
Line 252: remove “Now,”
Line 255 & 279: replace “speed” with “rate of reaction”
Line 257: “To be able to compare all of these results in one graph a logarithmic scale needed to be used (Figure 6).”
The difference in scale of the x-axis in Fig 5a/c and 5e shows the faster reaction with EDC/NHS compared to BDDGE and genipin rather clearly. Not sure all these graphs are needed in Fig 6 and 7. Maybe keep one and move the rest to SI.
Line 270 – Fig. 7: My impression is that this is giving the same information as Fig 6b. One Figure could move to SI, the analysis in lines 271-277 could stay.
Line 275: “…has a stronger of t0.85 vs. C…” – a word seems to be missing after “stronger”
Lines 288-298: a schematic with indicative chemical structures would add value here.
Line 293: “Now, let us add…” – please amend
Lines 297-298: not clear to me what “…even if after parting ways with polycaprolactone they remain crosslinked to each other.” refers to.
Line 299 – Section 3.3:
There is scope to present SEM images before and after cross-linking for the same cross-linking approach, to show indeed if the cross-linking process affects the fibre diameter in a negative way. We are bound to see some deterioration in the appearance & structure of fibres after the biodegradation test.
Lines 317-325: From this section it is unclear if it was expected that the high (> 85%) residual gelatin content would directly correlate to undamaged fibres, and if that is related to the cross-linking method.
Conclusions
Line 334: cost efficiency would also depend on the unit cost (e.g. EUR/g) of the cross-linkers. EDC/NHS may be added in lower concentrations, yet if these would be significantly more expensive per gram than the others, this may have an impact. Moreover, there are two components in EDC/NHS. The authors could provide a comparison of unit cost for each cross-linker and comment accordingly (see also comments in Table 1).
Lines 339-341: EDC/NHS is already a well-known system for this purpose, as the authors also mention. Perhaps a link to the authors' alternative solvent system and a related novelty in this direction would be of value to establish.
Reviewer 3 Report
The authors performed a systematic analysis of gelatin crosslinking within PCL/gelatin fibres, in order to optimize agents and parameters of the process, aiming to develop PCL/gelatin fibers with high gelatin content and stable morphology, applying a cost and time effective approach.
Although this issue is not new, this study is worth by the well-defined and systematic methodology, being useful for researchers with interest in this issue.
As PCL/gelatin fibers are intended for regenerative applications, the main limitation of this work is the absence of cytotoxicity/cytocompatibility studies. It is not known how the prepared fibers behave in the aqueous culture media and how they allow for cell growth. Even very basic assays, i.e. viability assay and SEM images of the colonized samples, would nicely complete this study.
Round 2
Reviewer 2 Report
The authors responded to the comments in a meticulous manner. Some additional points are given below.
FTIR spectroscopy: as long as the authors have obtained such data, I would suggest to present them with appropriate analysis in a supplementary information section (not in the main paper). I would accept the authors’ comment that it may be difficult to quantify and assess gelatin content by FTIR, but this will in fact strengthen the selection of their weight difference approach.
I would therefore suggest to add one paragraph in the Results section along the lines of the authors’ response to cover the fact that FTIR data were obtained but have certain limitations and hence are not shown in the main paper, and to present some characteristic spectra with some analysis/band assignment/any quantification in a supplementary information section.
Lines 320-324: added text appears to be underlined
Line 321: probably best to change to “The differences between these pairs of images for each crosslinking method are minimal…”
Line 135: “SEM imaging was performed…”
Lines 341-342: I think that the two occurrences of “how” in 341 and “were” in 342 can be omitted.
Author Response
Review (round 2)
The authors responded to the comments in a meticulous manner. Some additional points are given below.
FTIR spectroscopy: as long as the authors have obtained such data, I would suggest to present them with appropriate analysis in a supplementary information section (not in the main paper). I would accept the authors’ comment that it may be difficult to quantify and assess gelatin content by FTIR, but this will in fact strengthen the selection of their weight difference approach.
I would therefore suggest to add one paragraph in the Results section along the lines of the authors’ response to cover the fact that FTIR data were obtained but have certain limitations and hence are not shown in the main paper, and to present some characteristic spectra with some analysis/band assignment/any quantification in a supplementary information section.
The answer:
Thank you for your comments.
After reconsideration of FTIR results we decided to include a comparison of the spectra obtained for crosslinked materials after biodegradation test in the main paper. We think that our commentary of these results is better explained this way, rather than putting the spectra comparison in supplementary information section. So, there are two essential changes related to FTIR. The first one is the description of the FTIR method as a point 2.6 (rows 138-141) and the second one is description of FTIR results including one new figure (Fig. 10) and reference [29] as point 3.4 (rows 338-356). The samples analysed by FTIR shown in Fig. 10 correspond to samples shown on SEM images (Figs. 8, 9). Changes were indicated in cyan.
Lines 320-324: added text appears to be underlined
Text edited.
Line 321: probably best to change to “The differences between these pairs of images for each crosslinking method are minimal…”
Sentence corrected.
Line 135: “SEM imaging was performed…”
Missing word added.
Lines 341-342: I think that the two occurrences of “how” in 341 and “were” in 342 can be omitted.
Both “how” and “were” deleted.
Reviewer 3 Report
The authors answer to my comment.
Author Response
Thank you for your acceptance of our revision